# Sketch3D: Style-Consistent Guidance for Sketch-to-3D Generation

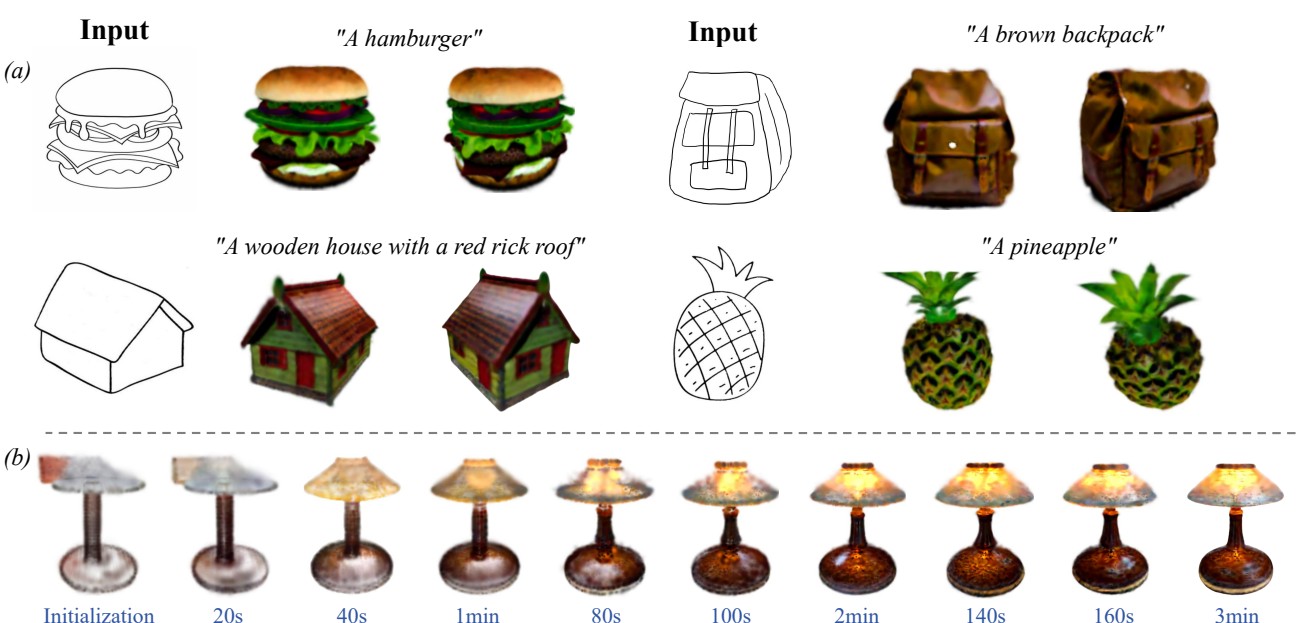

**Figure 1: Sketch3D aims at generating realistic 3D Gaussians with shape consistent with the input sketch and color aligned with textual description. (a) The novel-view generation results of four objects based on the input sketch and the text prompt. (b) Given a sketch of a lamp and text prompt "*A textural wooden lamp*", the 3D Gaussians progressively changes throughout the generation process. Our method can complete this generation process in about 3 minutes.**

## ABSTRACT

Recently, image-to-3D approaches have achieved significant results with a natural image as input. However, it is not always possible to access these enriched color input samples in practical applications, where only sketches are available. Existing sketch-to-3D researches suffer from limitations in broad applications due to the challenges of lacking color information and multi-view content. To overcome them, this paper proposes a novel generation paradigm **Sketch3D** to generate realistic 3D assets with shape aligned with the input sketch and color matching the textual description. Concretely, Sketch3D first instantiates the given sketch in the reference image through the shape-preserving generation process. Second, the reference image is leveraged to deduce a coarse 3D Gaussian prior, and multi-view style-consistent guidance images are generated based on the renderings of the 3D Gaussians. Finally, three strategies are designed to optimize 3D Gaussians, i.e., structural optimization via a distribution transfer mechanism, color optimization with a straightforward MSE loss and sketch similarity optimization with a CLIP-based geometric similarity loss. Extensive visual comparisons and quantitative analysis illustrate the advantage of our Sketch3D in generating realistic 3D assets while preserving consistency with the input.

## CCS CONCEPTS

• **Information systems** → *Multimedia content creation*.

## KEYWORDS

Sketch-to-3D Generation, 3D Gaussian Splatting, Diffusion Model

**Unpublished working draft. Not for distribution.**

## 1 INTRODUCTION

3D content generation is widely applied in various fields [16], including animation, movies, gaming, virtual reality, and industrial production. A 3D asset generative model is essential to enable non-professional users to easily transform their ideas into tangible 3D digital content. Significant efforts have been made to develop image-to-3D generation [10, 21, 53, 56], as it enables users to generate 3D

content based on color images. However, several practical scenarios provide only sketches as input due to the unavailability of colorful images. This is particularly true during the preliminary stages of 3D product design, where designers rely heavily on sketches. Despite their simplicity, these sketches are fundamental in capturing the core of the design. Therefore, it is crucial to generate realistic 3D assets according to the sketches.

Inspired by this practical demand, studies [14, 25, 40, 58] have endeavored to employ deep learning techniques in generating 3D shapes from sketches. Sketch2model [58] employs a view-aware generation architecture, enabling explicit conditioning of the generation process based on viewpoints. SketchSampler [6] proposed a sketch translator module to exploit the spatial information in a sketch and generate a 3D point cloud conforming to the shape of the sketch. Furthermore, recent works have explored the generation or editing of 3D assets containing color through sketches. Several [33, 50] proposed a sketch-guided method for colored point cloud generation, while others [19, 26] proposed a 3D editing technique to edit a NeRF based on input sketches. Despite these research advancements, there are still limitations hindering their widespread applications. **First**, generating 3D shapes from sketches typically lacks color information and requires training on extensive datasets. However, the trained models are often limited to generating shapes within a single category. **Second**, the 3D assets produced through sketch-guided generation or editing techniques often lack realism and the process is time-consuming.

These challenges inspire us to consider: *Is there a method to generate 3D assets where the shape aligns with the input sketch while the color corresponds to the textual description?* To address these shortcomings, we introduce **Sketch3D**, an innovative framework designed to produce lifelike 3D assets. These assets exhibit shapes that conform to input sketches while accurately matching colors described in the text. Concretely, a reference image is first generated via a shape-preserving image generation process. Then, we initialize a coarse 3D prior using 3D Gaussian Splatting [52], which comprises a rough geometric shape and a simple color. Subsequently, multi-view style-consistent guidance images can be generated using the IP-Adapter [51]. Finally, we propose three strategies to optimize 3D Gaussians: structural optimization with a distribution transfer mechanism, color optimization using a straightforward MSE loss and sketch similarity optimization with a CLIP-based geometric similarity loss. Specifically, the distribution transfer mechanism is employed within the SDS loss of the text-conditioned diffusion model, enabling the optimization process to integrate both the sketch and text information effectively. Furthermore, we formulate a reasonable camera viewpoint strategy to enhance color details via the $\ell_2$-norm loss function. Additionally, we compute the $L2$ distance between the mid-level activations of CLIP. As Figure 1 shows, our Sketch3D provides visualization results consistent with the input sketch and the textual description in just 3 minutes. These assets are readily integrable into software such as Unreal Engine and Unity, facilitating rapid application deployment.

To assess the performance of our method on sketches and inspire future research, we collect a ShapeNet-Sketch3D dataset based on the ShapeNet dataset [3]. Considerable experiments and analysis validate the effectiveness of our framework in generating 3D assets that maintain geometric consistency with the input sketch, while the color aligns with the textual description. Our contributions can be summarized as follows:

- We propose *Sketch3D*, a novel framework to generate realistic 3D assets with shape aligned to the input sketch and color matching the text prompt. To the best of our knowledge, this is the first attempt to steer the process of sketch-to-3D generation using a text prompt with 3D Gaussian splatting. Additionally, we have developed a dataset, named ShapeNet-Sketch3D, specifically tailored for research on sketch-to-3D tasks.
- We leverage IP-Adapter to generate multi-view style-consistent images and three optimization strategies are designed: a structural optimization using a distribution transfer mechanism, a color optimization with $\ell_2$-norm loss function, and a sketch similarity optimization using CLIP geometric similarity loss.
- Extensive qualitative and quantitative experiments demonstrate that our Sketch3D not only has convincing appearances and shapes but also accurately conforms to the given sketch image and text prompt.

## 2 RELATED WORK

### 2.1 Text-to-3D Generation

Text-to-3D generation aims at generating 3D assets from a text prompt. Recent developments in text-to-image methods [37–39] have demonstrated a remarkable capability to generate high-quality and creative images from given text prompts. Transferring it to 3D generation presents non-trivial challenges, primarily due to the difficulty in curating extensive and diverse 3D datasets. Existing 3D diffusion models [7, 9, 12, 24, 29, 31, 55, 60] typically focus on a limited number of object categories and face challenges in generating realistic 3D assets. To accomplish generalizable 3D generation, innovative works like DreamFusion [32] and SJC [48] utilize pre-trained 2D diffusion models for text-to-3D generation and demonstrate impressive results. Following works continue to enhance various aspects such as generation fidelity and efficiency [4, 11, 18, 20, 44, 46, 49, 54, 62], and explore further applications [1, 36, 42, 63]. However, the generated contents of text-to-3D method are unpredictable and the shape cannot be controlled according to user requirements.

### 2.2 Sketch-to-3D Generation

Sketch-to-3D generation aims to generate 3D assets from a sketch image and possible text input. Since sketches are highly abstract and lack substantial information [41], generating 3D assets based on sketches becomes a challenging problem. Sketch2Model [58] introduces an architecture for view-aware generation that explicitly conditions the generation process on specific viewpoints. Sketch2Mesh [8] employs an encoder-decoder architecture to represent and adjust a 3D shape so that it aligns with the target external contour using a differentiable renderer. SketchSampler [6] proposes a sketch translator module to utilize the spatial information within a sketch and generate a 3D point cloud that represents the shape of the sketch. Sketch-A-Shape [40] proposes a zero-shot approach for sketch-to-3D generation, leveraging large-scale pre-trained models.

SketchFaceNeRF [19] proposes a sketch-based 3D facial NeRF generation and editing method. SKED [26] proposes a sketch-guided 3D editing technique to edit a NeRF. Overall, existing sketch-to-3D generation methods have several limitations. First, generating 3D shapes from sketches invariably produces shapes without color information and needs to be trained on large-scale datasets, yet the trained models are typically limited to making predictions on a single category. Second, the 3D assets generated by sketch-guided generation or editing techniques often lack realism, and the process is relatively time-consuming. Our method, incorporating the input text prompt, is capable of generating 3D assets with shapes consistent with the sketch and color aligned with the textual description.

## 3 METHOD

In this section, we first introduce two preliminaries including 3D Gaussian Splatting and Controllable Image Synthesis (Sec. 3.1). Subsequently, we systematically propose our **Sketch3D** framework (Sec. 3.2), which is progressively introduced (Sec. 3.3–3.5).

### 3.1 Preliminaries

**3D Gaussian Splatting.** 3D Gaussian Splatting (3DGS) [13] represents a novel method for novel-view synthesis and 3D scene reconstruction, achieving promising results in both quality and real-time processing speed. Unlike implicit representation methods such as NeRF [27], 3D Gaussians represent the scene through a set of anisotropic Gaussians, defined with its center position $\mu \in \mathbb{R}^3$, covariance $\Sigma \in \mathbb{R}^7$, color $\mathbf{c} \in \mathbb{R}^3$, and opacity $\alpha \in \mathbb{R}^1$. The covariance matrix $\Sigma = \mathbf{R}\mathbf{S}\mathbf{S}^\top\mathbf{R}^\top$ describes the configuration of an ellipsoid and is implemented via a scaling matrix $\mathbf{S}$ and a rotation matrix $\mathbf{R}$. Each Gaussian centered at point (mean) $\mu$ is defined as:

$$G(x) = e^{-\frac{1}{2}x^\top \Sigma^{-1} x}, \tag{1}$$

where $x$ represents the distance between $\mu$ and the query point. A ray $r$ is cast from the center of the camera, and the color and density of the 3D Gaussians that the ray intersects are computed along the ray. In summary, $G(x)$ is multiplied by $\alpha$ in the blending process to construct the final accumulated color:

$$C(r) = \sum_{i=1}^{N} c_i \alpha_i G(x_i) \prod_{j=1}^{i-1} \left(1 - \alpha_j G(x_j)\right), \tag{2}$$

where $N$ means the number of samples on the ray $r$, $c_i$ and $\alpha_i$ denote the color and opacity of the $i$-th Gaussian.

**Controllable Image Synthesis.** In the field of image generation, achieving control over the output remains a great challenge. Recent efforts [15, 17, 59] have focused on increasing the controllability of generated images by various methods. This involves increasing the ability to specify various attributes of the generated images, such as shape and style. ControlNet [57] and T2I-adapter [28] attempt to control image creation utilizing data from different visual modalities. Specifically, ControlNet is an end-to-end neural network architecture that controls a diffusion model (Stable Diffusion [38]) to adapt task-specific input conditions. IP-Adapter [51] and MasaCtrl [2] leverage the attention layer to incorporate information from additional images, thus achieving enhanced controllability over the generated results.

### 3.2 Framework Overview

Given a sketch image and a corresponding text prompt, our objective is to generate realistic 3D assets that align with the shape of the sketch and correspond to the color described in the textual description. To achieve this, we confront three challenges:

- *How to solve the problem of missing information in sketches?*
- *How to initialize a valid 3D prior from an image?*
- *How to optimize 3D Gaussians to be consistent with the given sketch and the text prompt?*

Inspired by this motivation, we introduce a novel 3D generation paradigm, named *Sketch3D*, comprising three dedicated steps to tackle each challenge (as illustrated in Figure 2):

- **Step 1**: Generate a reference image based on the input sketch and text prompt (Sec. 3.3).
- **Step 2**: Derive a coarse 3D prior using 3D Gaussian Splatting from the reference image (Sec. 3.4).
- **Step 3**: Generate multi-view style-consistent guidance images through IP-Adapter, introducing three strategies to facilitate the optimization process (Sec. 3.5).

### 3.3 Shape-Preserving Reference Image Generation

For image-to-3D generation, sketches offer very limited information, when served as a visual prompt compared with RGB images. They lack color, depth, semantic information, etc., and only contain simple contours.

To solve the above problems, our solution is to create a shape-preserving reference image from a sketch $I_s$ and a text prompt $y$. The reference image adheres to the outline of the sketch, while also conforming to the textual description. To achieve this, we leverage an additional image conditioned diffusion model $G_{2D}$ (e.g., ControlNet [57]) to initiate sketch-preserving image synthesis [5]. Given time step $t$, a text prompt $y$, and a sketch image $I_s$, $G_{2D}$ learn a network $\hat{\epsilon}_\theta$ to predict the noise added to the noisy image $x_t$ with:

$$\mathcal{L} = \mathbb{E}_{x_0, t, y, I_s, \epsilon \sim \mathcal{N}(0,1)} \left[ \|\hat{\epsilon}_\theta(x_t; t, y, I_s) - \epsilon\|_2^2 \right], \tag{3}$$

where $\mathcal{L}$ is the overall learning objective of $G_{2D}$. Note that there are two conditions, i.e., sketch $I_s$ and text prompt $y$, and the noise is estimated as follows:

$$\begin{aligned} \hat{\epsilon}_\theta(x_t; t, y, I_s) =& \epsilon_\theta(x_t; t) \\ & + w * (\epsilon_\theta(x_t; t, y, I_s) - \epsilon_\theta(x_t; t)), \end{aligned} \tag{4}$$

where $w$ is the scale of classifier-free guidance [30]. In summary, $G_{2D}$ can quickly generate a shape-preserving colorful image $I_{ref}$ that not only follows the sketch outline but also respects the textual description, which facilitates the subsequent initialization process.

### 3.4 Gaussian Representation Initialization

A coarse 3D prior can efficiently offer a solid initial basis for subsequent optimization. To facilitate image-to-3D generation, most existing methods rely on implicit 3D representations such as Neural Radiance Fields (NeRF) [45] or explicit 3D representations such as mesh [34]. However, NeRF representations are time-consuming and require high computational resources, while mesh representations have complex representational elements. Consequently, 3D

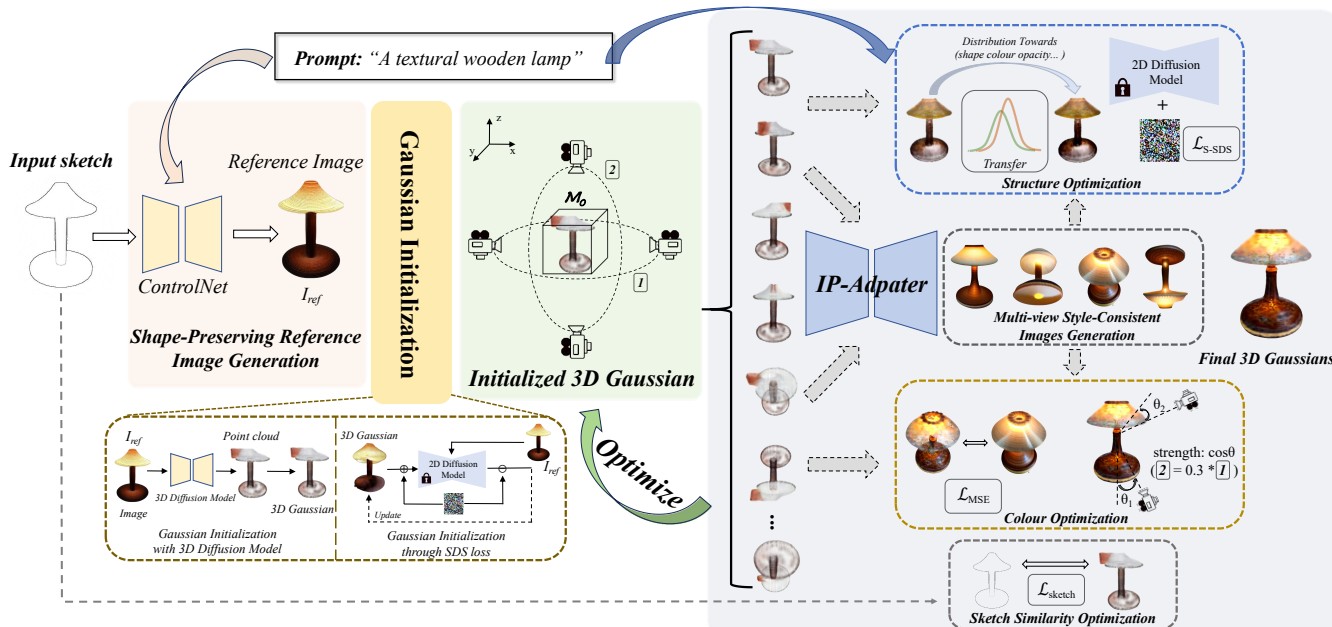

**Figure 2: Pipeline of our Sketch3D. Given a sketch image and a text prompt as input, we first generate a reference image $I_{\text{ref}}$ using ControlNet. Second, we utilize the reference image $I_{\text{ref}}$ to initialize a coarse 3D prior $M_0$, which is represented using 3D Gaussians. Third, we render the 3D Gaussians into images from different viewpoints using a designated camera projection strategy. Based on these, we obtain multi-view style-consistent guidance images through the IP-Adapter. Finally, we formulate three strategies to optimize $M_0$: (a) Structural Optimization: a distribution transfer mechanism is proposed for structural optimization, effectively steering the structure generation process towards alignment with the sketch. (b) Color Optimization: based on multi-view style-consistent images, we optimize color with a straightforward MSE loss. (c) Sketch Similarity Optimization: a CLIP-based geometric similarity loss is used as a constraint to shape towards the input sketch.**

Gaussian representation, being simple and fast, is chosen as our initialized 3D prior $M_0$.

**Gaussian Initialization with 3D Diffusion Model.** 3D Gaussians can be easily converted from a point cloud, so a simple idea is to first obtain an initial point cloud and then convert it to 3D Gaussians [52]. Therefore, it can be transformed into an image-to-point cloud problem. Currently, many 3D diffusion models use text to generate 3D point clouds [29]. However, we initialize 3D Gaussians $M_0$ from 3D diffusion model $G_{3D}$ (e.g., Shap-E [12]) based on the image $I_{\text{ref}}$.

**Gaussian Initialization through SDS loss.** Alternatively, we can also initialize a Gaussian sphere and optimize it into a coarse Gaussian representation through SDS loss [44]. First, we initialize the 3D Gaussians with random positions sampled inside a sphere, with unit scaling and no rotation. At each step, we sample a random camera pose $p$ orbiting the object center and render the RGB image $I_{\text{RGB}}^p$ of the current view. Stable-zero123 [43] is adopted as the 2D diffusion prior $\phi$ and the images $I_{\text{RGB}}^p$ are given as input. The SDS loss is formulated as:

$$\nabla_\theta \mathcal{L}_{\text{SDS}} = \mathbb{E}_{t,p,\epsilon}\left[\left(\epsilon_\phi\left(I_{\text{RGB}}^p; t, I_{\text{ref}}, \Delta p\right) - \epsilon\right)\frac{\partial I_{\text{RGB}}^p}{\partial \theta}\right], \quad (5)$$

where $\epsilon_\phi(\cdot)$ is the predicted noise by the 2D diffusion prior $\phi$, and $\Delta p$ is the relative camera pose change from the reference camera. Finally, we can obtain a coarse Gaussian representation $M_0$ based on the optimization of the 2D diffusion prior $\phi$.

### 3.5 Style-Consistent Guidance for Optimization

The coarse 3D prior $M_0$ is roughly similar in shape to the input sketch, and its color is not completely consistent with the text description. Specifically, the geometric shape generated in Sec. 3.4 may not exactly fit the outline shape of the input sketch $I_s$, and there is a certain deviation. For example, the input sketch is an upright, cylinder-like, symmetrical lamp, but the coarse 3D Gaussian representation might be a slightly curved, asymmetrical lamp. Moreover, the initial color generated in Sec. 3.4 may not be consistent with the description of the input text. Faced with these problems, we introduce IP-Adapter to generate multi-view style-consistent images as guidance. First, we propose a transfer mechanism in the structural optimization process, which can effectively guide the structure of the 3D Gaussian representation to align with the input sketch outline. Second, we utilize a straightforward MSE loss to improve the color quality, which can effectively align the 3D Gaussian representation with the input text description. Third, we implement a CLIP-based geometric similarity loss as a constraint to guide the shape towards the input sketch.

**Multi-view Style-Consistent Images Generation.** Due to the rapid and real-time capabilities of Gaussian splatting, acquiring multi-view renderings becomes straightforward. If we can obtain guidance images from these renderings, corresponding to the current viewing angles, they would serve as effective guides for

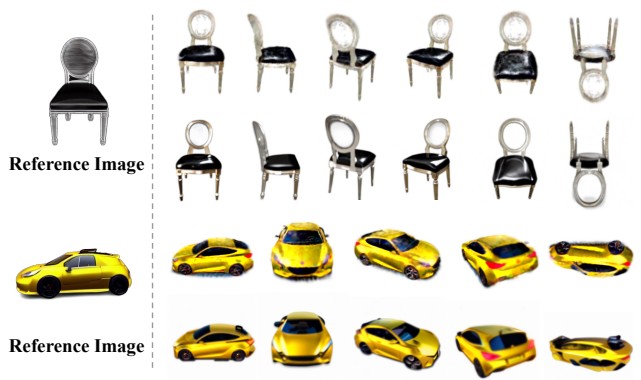

**Figure 3: For each object, the first row shows content images and the second row shows guidance images. Given reference image $I_{ref}$ generated by ControlNet and content images $I_c$ rendered from the 3D Gaussians, we generate the guidance images $I_g$ as the multi-view style-consistent images.**

optimization. To achieve this, we introduce the IP-Adapter [51], which incorporates an additional cross-attention layer for each cross-attention layer in the original U-Net model to insert image features. Given the image features $c_i$, the output of additional cross-attention $\mathbf{Z}$ is computed as follows:

$$\mathbf{Z} = \text{Attention}(\mathbf{Q}, \mathbf{K}_i, \mathbf{V}_i) = \text{Softmax}\left(\frac{\mathbf{Q}\mathbf{K}_i^\top}{\sqrt{d}}\right)\mathbf{V}_i, \qquad (6)$$

where $\mathbf{Q} = \mathbf{Z}\mathbf{W_q}$, $\mathbf{K}_i = c_i\mathbf{W}'_k$ and $\mathbf{V}_i = c_i\mathbf{W}'_v$ are the query, key, and values matrices from the image features. $\mathbf{Z}$ is the query features, and $\mathbf{W}'_k$ and $\mathbf{W}'_v$ are the corresponding trainable weight matrices.

This enhancement enables us to generate multi-view style-consistent images based on the two image conditions of the reference image and the content images, as shown in Figure 3. Specifically, herein we use Stable-Diffusion-v1-5 [38] as our diffusion model basis. Given the reference image $I_{ref}$ and multi-view splatting images as the content images $I_c$, the guidance images $I_g$ are estimated as follows:

$$I_g = M(I_{ref}, I_c, t, \lambda), \qquad (7)$$

where $M$ is the generator of IP-Adapter, $t$ is the sampling time step of inference, and $\lambda \in [0, 1]$ is a hyper-parameter that determines the control strength of the conditioned content image $I_c$.

**Camera Projection Strategy.** As shown in Figure 2, during the process of Gaussian splatting, our camera projection strategy involves encircling horizontally and vertically. To ensure the stylistic consistency of the generated guidance images, we perform a rotation every 30 degrees for each circle, thereby calculating the guidance images under a progressively changing viewpoint.

**Structural Optimization.** For image-to-3D generation, when selecting the diffusion prior for SDS loss, existing approaches usually use a diffusion model with image as a condition (e.g., Zero-123 [23]). Differently, we use a diffusion model with text as the condition (e.g., Stable Diffusion [38]). The reason is that the former does not perform well in generating 3D aspects of the invisible parts of the input image while the latter demonstrates better optimization effects in terms of details and the invisible sections. However, we have to ensure that the reference image plays an important role in

the optimization process, so we propose a mechanism of distribution transfer and then use it in subsequent SDS loss calculations. Given guidance images $I_g$ and splatting images $I_c$, the transferred images $I_t$ are estimated as follows:

$$I_t = \sigma\left(I_g\right)\left(\frac{I_c - \mu(I_c)}{\sigma(I_c)}\right) + \mu\left(I_g\right), \qquad (8)$$

where $\mu(\cdot)$ is the mean operation, $\sigma(\cdot)$ is the variance operation. The distribution transformation brought about by the transfer mechanism can bring the distribution of splatting images closer to the distribution of guidance images. In this way, we obtain the transfer image $I_t$ after distribution migration through guidance image $I_g$ and splatting image $I_c$. To update the 3D Gaussian parameters $\theta\ (\mu, \Sigma, c, \alpha)$, we choose to use the publicly available *Stable Diffusion* [38] as 2D diffusion model prior $\phi$ and compute the gradient of the SDS loss via:

$$\nabla_\theta \mathcal{L}_{S-SDS} = \mathbb{E}_{t,\epsilon}\left[(\hat{\epsilon}_\phi(I_t; t, y, I_s) - \epsilon)\frac{\partial I_t}{\partial \theta}\right], \qquad (9)$$

where $I_t$ is the transfer image, $y$ is text prompt, $\hat{\epsilon}_\phi$ is similar to Equation 4, $t$ is the sampling time step, and $I_s$ is the input sketch. In conclusion, through the tailored 3D structural guidance, our Sketch3D can mitigate the problem of geometric inconsistencies.

**Color Optimization.** Although through the above structural optimization, we already obtained a 3D Gaussian representation whose geometric structure is highly aligned with the input sketch, some color details still need to be enhanced. To improve the image color quality, we propose to use a simple MSE loss to optimize the 3D Gaussian parameters $\theta$. We optimize the splatting image $I_c$ to align with the guidance image $I_g$.

$$\mathcal{L}_{Col} = \lambda_{pose} * \lambda_{linear}||I_g - I_c||_2^2, \qquad (10)$$

where $\lambda_{linear}$ is the linearly increased weight during optimization, calculated by dividing the current step by the total number of iteration steps. $I_g$ represents the guidance images obtained from controllable IP-Adapter and $I_c$ represents the splatting images from 3D Gaussian. The MSE loss is fast to compute and deterministic to optimize, resulting in fast refinement. Note that $\lambda_{pose}$ is a parameter that changes with viewing angle, as shown in Figure 2, in the horizontal rotation perspective, the value of $\lambda_{pose}$ is $\cos(\theta_{azimuth})$, in the vertical rotation perspective, the value of $\lambda_{pose}$ is $0.3 * \cos(\theta_{elevation})$.

**Sketch Similarity Optimization.** To ensure that the shape of the sketch can directly guide the optimization of 3D Gaussians, we use the image encoder of CLIP to encode both the sketch and the rendered images, and compute the $L2$ distance between intermediate level activations of CLIP. CLIP is trained on various image modalities, enabling it to encode information from both images and sketches, without requiring further training. CLIP encodes high-level semantic attributes in the last layer since it was trained on both images and text. One intuitive approach involves leveraging CLIP's semantic-level cosine similarity loss to use the sketch as a supervisory signal for the shape of rendered images. However, this form of supervision is quite weak. Therefore, to measure an effective geometric similarity loss between the sketch and rendered image, ensuring that the shape of rendered images is more consistent with the input sketch, we compute the $L2$ distance between

the mid-level activations [47] of CLIP:

$$\mathcal{L}_{\text{sketch}} = \lambda_{\text{sketch}} * \|CLIP_4(I_s) - CLIP_4(I_c)\|_2^2, \quad (11)$$

where $\lambda_{\text{sketch}}$ is a coefficient that controls the weight, $CLIP_4(\cdot)$ is the *CLIP* encoder activation at layer 4. Specifically, we use layer 4 of the ResNet101 CLIP model.

## 4 EXPERIMENTS

In this section, we first introduce the experiment setup in Sec. 4.1, then present qualitative visual results compared with five baselines and report quantitative results in Sec. 4.2. Finally, we carry out ablation and analytical studies to further verify the efficacy of our framework in Sec. 4.3.

### 4.1 Experiment Setup

**ShapeNet-Sketch3D Dataset.** To evaluate the effectiveness of our method and benefit further research, we have collected a comprehensive dataset comprising 3D objects, synthetic sketches, rendered images, and corresponding textual descriptions, which we call ShapeNet-Sketch3D. It contains object renderings of 10 categories from ShapeNet [3], and there are 1100 objects in each category. Rendered images from 20 different views of each object are rendered in $512 \times 512$ resolution. We extract the edge map of each rendered image using a canny edge detector. The textual descriptions corresponding to each object were derived by posing questions to GPT-4-vision about their rendered images, leveraging its advanced capabilities in visual analysis. Currently, there are no datasets available for paired sketches, rendered images, textual descriptions, and 3D objects. Our dataset serves as a valuable resource for research and experimental validation in sketch-to-3D tasks.

**Implementation Details.** In the shape-preserving reference image generation process, we use control-v11p-sd15-canny [57] as our diffusion model $G_{2D}$. In the Gaussian initialization process, we initialize our Gaussian representation with the 3D diffusion model and utilize Shap-E [12] as our 3D diffusion model $G_{3D}$. In the multi-view style-consistent image generation process, we use the stable diffusion image-to-image pipeline [51], with a control strength of 0.5. Moreover, we generate two sets of guidance images in two surround modes every 30 steps. In structural optimization, we use stablediffusion-2-1-base [38]. The total training steps are 500. For the 3D Gaussians, the learning rates of position $\mu$ and opacity $\alpha$ are $10^{-4}$ and $5 \times 10^{-2}$. The color $c$ of the 3D Gaussians is represented by the spherical harmonics (SH) coefficient, with a learning rate of $1.5 \times 10^{-2}$. The covariance of the 3D Gaussians is converted into scaling and rotation for optimization, with learning rates of $5 \times 10^{-3}$ and $10^{-3}$. We select a fixed camera radius of 3.0, y-axis FOV of 50 degree, with the azimuth in [0, 360] degrees and elevation in [0, 360] degrees. The rendering resolution is $512 \times 512$ for Gaussian splatting. All our experiments can be completed within 3 minutes on a single NVIDIA RTX 4090 GPU with a batch size of 4.

**Baselines.** We extensively compare our method Sketch3D against five baselines: Sketch2Model [58], LAS-Diffusion [61], Shap-E [12], One-2-3-45 [22], and DreamGaussian [44]. We do not compare with NeRF-based methods, as they typically require a longer time to generate. Sketch2Model is the pioneering method that explores the generation of 3D meshes from sketches and introduces viewpoint

**Table 1: Quantitative comparisons on CLIP similarity and Structural Similarity Index Measure (SSIM) with other methods. All these experiments were conducted on our ShapeNet-Sketch3D dataset.**

| Method | CLIP-Similarity | | SSIM ↑ |
|---|---|---|---|
| | pic2pic ↑ | pic2text ↑ | |
| Sketch2Model | 0.597 | 0.232 | 0.712 |
| LAS-Diffusion | 0.638 | 0.254 | 0.731 |
| Shap-E | 0.642 | 0.268 | 0.734 |
| One-2-3-45 | 0.667 | 0.281 | 0.722 |
| DreamGaussian | 0.724 | 0.294 | 0.793 |
| **Sketch3D (Ours)** | **0.779** | **0.321** | **0.818** |

judgment to optimize shapes. LAS-Diffusion leverages a view-aware local attention mechanism for image-conditioned 3D shape generation, utilizing both 2D image patch features and the SDF representation to guide the learning of 3D voxel features. Shap-E is capable of generating 3D assets in a short time, but requires extensive training on large-scale 3D datasets. One-2-3-45 employs Zero123 to generate results of the input image from different viewpoints, enabling the rapid creation of a 3D mesh from an image. DreamGaussian integrates 3D Gaussian Splatting into 3D generation and greatly improves the speed.

### 4.2 Comparisons

**Qualitative Comparisons.** Figure 4 displays the qualitative comparison results between our method and the five baselines, while Figure 1 shows novel-view images generated by our method. Sketch3D achieves the best visual results in terms of shape consistency and color generation quality. As illustrated in Figure 4, the sketch image and the reference image generated in Section 3.3 are chosen as inputs for the latter three baselines. For the same object, the reference image used by the latter three baselines and our method is identical. **First**, Sketch2Model and LAS-Diffusion only generate shapes and lack color information. **Second**, Shap-E can generate a rough shape and simple color, but the color details are blurry. **Third**, One-2-3-45 and DreamGaussian often produce inconsistent shapes and lack color details. All of these results demonstrate the superiority of our method. Additionally, Sketch3D is capable of generating realistic 3D objects in about 3 minutes.

**Quantitative Comparisons.** In Table 1 , we use CLIP similarity [35] and structural similarity index measure (SSIM) to quantitatively evaluate our method. We randomly select 5 objects from each category in our ShapeNet-Sketch3D dataset, choose a random viewpoint for each object, and then average the results across all objects. We calculate the CLIP similarity between the final rendered images and the reference image, as well as between the final rendered images and the text prompt. Moreover, we also calculate the SSIM similarity between the final rendered images and the reference image. The results show that our method can better align with the input sketch shape and correspond to the input textual description.

### 4.3 Ablation Study and Analysis

**Distribution transfer mechanism in structural optimization.** As shown in Figure 5, the distribution transfer mechanism aligns

| Input | Sketch2Model | LAS-Diffusion | Shap-E | One-2-3-45 | DreamGaussian | **Ours** |
|-------|--------------|---------------|--------|------------|---------------|----------|

*"A black and wooden chair"*

*"A golden sports car"*

*"A colourful garden bench"*

*"A textural wooden lamp"*

*"A blue modern plane"*

*"A small oak round table"*

*"A red fresh cherry"*

*"A banana"*

**Figure 4: Qualitative comparisons between our method and Sketch2Model [58], LAS-Diffusion [61], Shap-E [12], One-2-3-45 [22] and DreamGaussian [44]. The input sketches include sketch images, exterior contour sketches, and hand-drawn sketches. Our method achieves the best visual results regarding shape consistency and color generation quality compared to other methods.**

the shape more closely with the input sketch, leading to a coherent structure and color. It demonstrates the mechanism's effectiveness in steering the generated shape towards the input sketch.

**MSE loss in color optimization.** As illustrated in Figure 5, it is evident that the MSE loss contributes to reducing color noise, leading to a smoother overall color appearance. It proves that MSE loss is capable of enhancing the quality of the generated color.

**CLIP geometric similarity loss in sketch similarity optimization.** As shown in Figure 5, the CLIP geometric similarity loss enables the overall shape to more closely align with the shape of

the input sketch. This illustrates that the $L2$ loss in the intermediate layers of CLIP can act as a shape constraint.

**Gaussian initialization through SDS loss.** As shown in Figure 6, we conducted analytical experiments on the Gaussian initialization method to explore which initialization method is better. It can be seen that Gaussian initialization through SDS loss shows good 3D effects only in the visible parts of the input reference image, while problems of blurriness and color saturation exist in the invisible parts. However, the approach of Gaussian initialization with a 3D diffusion model exhibits better realism from all viewing angles.

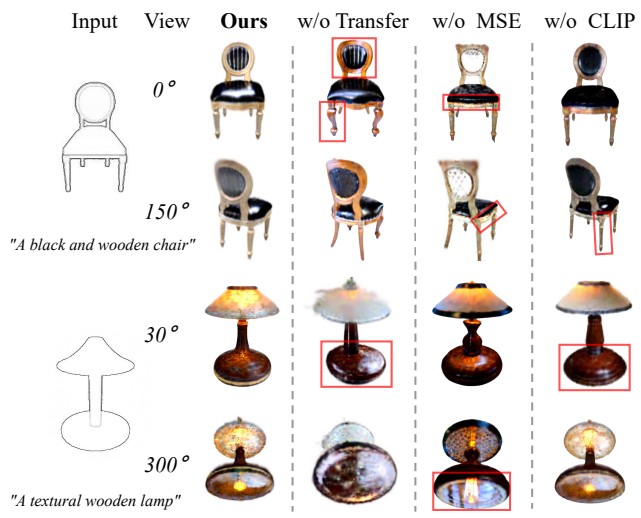

Figure 5: Ablation study. Two different angles are selected for each object. Red boxes show details.

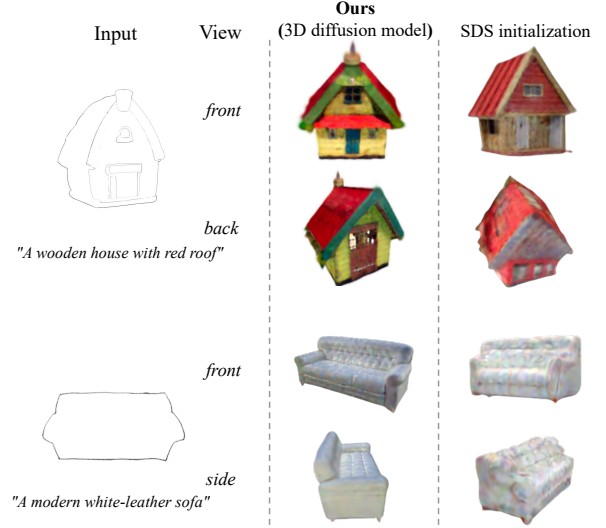

Figure 6: Analytical study of the initialization approach of the 3D Gaussian Representation.

**Hand-drawn sketch visualization results.** As shown in Figure 7, to explore the fidelity of outcomes generated from the user's freehand sketches, we visualize some of the generated results from hand-drawn sketches. We randomly select three non-artist users to draw three sketches and provide corresponding text prompts. The results show that our method can also achieve good generation quality and consistency for hand-drawn sketches.

**User Study.** We additionally conduct a user study to quantitatively evaluate Sketch3D against four baseline methods (LAS-Diffusion, Shap-E, One-2-3-45, and DreamGaussian). We invite 9 participants and present them with each input and the corresponding 5 generated video results, comprising a total of 10 inputs and the corresponding 50 videos. We ask each participant to rate each video on a scale from 1-5 based on fidelity and consistency criteria.

Table 2: User Study on fidelity and consistency evaluation.

| Method | Fidelity | Consistency |
|---|---|---|
| LAS-Diffusion | 1.82 | 2.86 |
| Shap-E | 2.67 | 2.92 |
| One-2-3-45 | 3.22 | 3.58 |
| DreamGaussian | 3.78 | 3.37 |
| **Sketch3D** | **4.12** | **3.94** |

Table 2 shows the results of the user study. Overall, our Sketch3D demonstrates greater fidelity and consistency than the other four baselines.

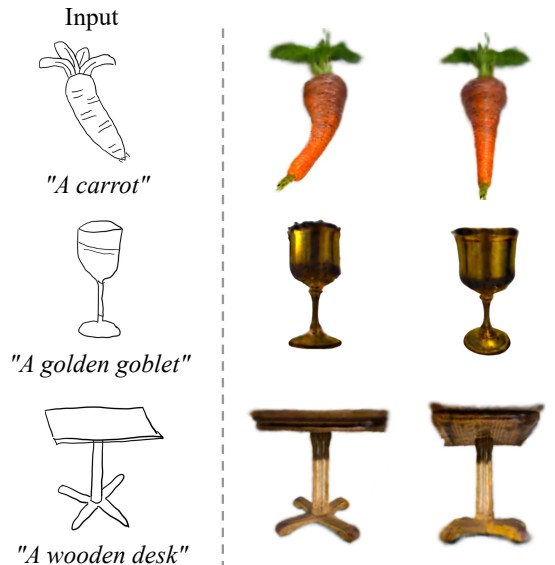

Figure 7: Hand-drawn sketch visualization results.

## 5 CONCLUSION

In this paper, we propose Sketch3D, a new framework to generate realistic 3D assets with shape aligned to the input sketch and color matching the text prompt. Specifically, we first instantiate the given sketch to the reference image through the shape-preserving generation process. Second, a coarse 3D Gaussian prior is sculpted based on the reference image, and multi-view style-consistent guidance images could be generated using IP-Adapter. Third, we propose three optimization strategies: a structural optimization using a distribution transfer mechanism, a color optimization using a straightforward MSE loss, and a sketch similarity optimization using CLIP geometric similarity loss. Extensive experiments demonstrate that Sketch3D not only has realistic appearances and shapes but also accurately conforms to the given sketch and text prompt. Our Sketch3D is the first attempt to steer the process of sketch-to-3D generation with 3D Gaussian splatting, providing a valuable foundation for future research on sketch-to-3D generation. However, our method also has several limitations. The quality of the reference image depends on the performance of ControlNet, so when the image quality generated by the ControlNet is poor, it will affect our method and impact the overall generation quality. Additionally, for particularly complex or richly detailed sketches, it is difficult to achieve control over the details in the output results.

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
