# OpenReview forum: "Sketch3D: Style-Consistent Guidance for Sketch-to-3D Generation"
_acmmm.org/ACMMM/2024/Conference — MM2024 Poster_

### Official Review · Reviewer_mfJT · 2024-05-09

**Rating:** 3
**Confidence:** 3

**Summary:**

This paper proposes a novel generation paradigm Sketch3D to generate realistic 3D assets with shape aligned with the input sketch and color matching the textual description. Their contributions can
be summarized as follows:
• Authors propose Sketch3D, a novel framework to generate realistic 3D assets with shape aligned to the input sketch and color matching the text prompt. To the best of our knowledge, this is the first attempt to steer the process of sketch-to-3D generation using a text prompt with 3D Gaussian splatting. Additionally, we have developed a dataset, named ShapeNet-Sketch3D, specifically tailored for research on sketch-to-3D tasks.
• Authors leverage IP-Adapter to generate multi-view style-consistent images and three optimization strategies are designed: a structural optimization using a distribution transfer mechanism, a color optimization with ℓ2-norm loss function, and a sketch similarity optimization using CLIP geometric similarity loss.
• Extensive qualitative and quantitative experiments demonstrate that their Sketch3D not only has convincing appearances and shapes but also accurately conforms to the given sketch image and text prompt.

**Strengths:**

1. Gaussian work brings new research potential to 3D generation. The author innovatively combines sketching with 3D generation tasks through Gaussian.
2. Smooth writing and clear logic
3. The illustrations are clear and easy to understand.

**Limitations:**

1. Although the author's effect looks good, I feel that there are some limitations in the 3D generation work of introducing Gaussian, and it does not realize the research potential of Gaussian. For example, the wings on both sides, the corners of the round table, and the cherry platycodon in Figure 4 are not very effective.
2. MSE should constrain the color so that the color of the generated 3D is similar to the color of the generated image. However, I did not see the outstanding effect of MSE. The color of the lampshade does not seem to be consistent with the color of the image. I also did not see the effect of MSE in the visualization results of ablation loss. If the author could provide me with more results, I might have a new perspective.
3. The author uses the diffusion model to generate images aligned with the sketch as constraints for 3D generation. I understand that the 3D model will definitely undergo some changes during the subsequent optimization process (Figure 6 and Figure 7 above), but I want to know if this change process is uncontrollable? (Points appeared above the house, wine glasses were uneven, table corners were blunted) I think I would be happy to adjust my rating if it was controllable.

**Suitability:**

2

---

### Official Review · Reviewer_oQ1g · 2024-05-24

**Rating:** 4
**Confidence:** 3

**Summary:**

The paper presents Sketch3D, a novel framework designed to generate realistic 3D assets based on input sketches and textual descriptions. The approach addresses the limitations of existing sketch-to-3D generation methods, which often lack color information and require extensive datasets. Sketch3D leverages 3D Gaussian Splatting, ControlNet for shape-preserving reference image generation, and IP-Adapter for multi-view style-consistent image generation. The framework incorporates three optimization strategies: structural optimization via distribution transfer, color optimization with MSE loss, and sketch similarity optimization using CLIP-based geometric similarity loss. The effectiveness of Sketch3D is demonstrated through extensive visual comparisons and quantitative analysis, showing superior performance in generating lifelike 3D assets consistent with input sketches and textual descriptions.

**Strengths:**

+ The combination of 3D Gaussian Splatting, ControlNet, and IP-Adapter introduces a novel way to generate 3D assets from sketches and text, addressing key limitations of previous methods.
+ Dataset Contribution: The introduction of the ShapeNet-Sketch3D dataset provides a valuable resource for future research in sketch-to-3D generation tasks.
Extensive Evaluation: The paper includes comprehensive experiments, ablation studies, and user studies to validate the proposed method, offering a thorough assessment of its strengths and limitations.

**Limitations:**

- Dependency on ControlNet: The quality of the reference images generated by ControlNet significantly impacts the overall performance of Sketch3D. Poor image quality from ControlNet can degrade the final 3D asset generation.
- Complexity in Detailed Sketches: For highly detailed or complex sketches, the method may struggle to maintain control over the finer details in the output, potentially reducing the fidelity of the generated assets.
- Limited Generalization: While the method performs well on the provided dataset, its generalization to more diverse and unstructured sketches or textual descriptions needs further exploration.
- Lack of Complex Text and Object Experiments: The paper lacks experimental results involving complex text and objects, such as "a fox", "a dog reading a book", and "an astronaut riding a horse". Including such examples would better demonstrate the model's capabilities and limitations in handling more intricate scenarios.

**Suitability:**

3

---

### Official Review · Reviewer_BbGS · 2024-05-25

**Rating:** 2
**Confidence:** 3

**Summary:**

The paper presents a sketch-text guided 3D content generation method to tackle the limitations of existing solutions in terms of fidelity and consistency. The proposed Sketch 3D first generates a reference image guided by a pair of sketch-text prompt for a coarse 3D prior estimation using Gaussian Splatting, and then refines the 3D content with three newly developed optimisation stategies. Both qualitative and quantitative experiments were carried out to demonstrate the effectiveness of the proposed method.

**Strengths:**

Overall, the idea introduced in this work is interesting and is one of the pioneering attempts that leverage multimodal prompts and recent advance in computer graphics to deal with the challenges in 3D AIGC.

**Limitations:**

However, important details about the experiments are missing, and this makes the conclusion less convincing.

In Sec. 4.1, the author(s) claim that they did not compare NeRF-based methods since they are typically time-consuming. It would be interesting to see comparisons  between NeRF-based methods and the proposed one with regard to both quality and efficiency.

In Sec. 4.3, a user study was conducted to evaluate the superiority of the proposed method against four baseline methods. Unfortunately, the background of the 9 participants involved in the study and more details about the evaluation process are missing. Without sufficient details, the results in Table 2 are not convincing.

**Suitability:**

3

---

### Official Review · Reviewer_YAWB · 2024-05-30

**Rating:** 5
**Confidence:** 3

**Summary:**

Sketch3D has developed a pioneering framework designed to create realistic 3D assets. This framework aligns the shape of the generated 3D assets with an input sketch, while also ensuring the color matches a provided text prompt. Furthermore, the researchers have compiled a specialized dataset, ShapeNet-Sketch3D, which will facilitate research in the sketch-to-3D domain.

**Strengths:**

- the first attempt to steer the process of sketch-to-3D generation using a text prompt with 3D Gaussian splatting.
- 3D Gaussian initialization approach is resonable.
- reasonable optimization including structural optimization, color optimization and sketch similarity optimization.
- good results.

**Limitations:**

- Table1. How do you get the rendered images for Sketch2Model and LAS-Diffusion which lack textures?
- L630: does 3min include the initialization?
- Figure 6 shows initialization results of different results, could you please show the final results of these objects?

**Suitability:**

3

---

### Meta-Review · Area_Chair_wfCn · 2024-07-01

**Recommendation:** Accept (Poster)
**Confidence:** 5

**Metareview:**

This paper presents a new model, Sketch3D, designed for the task of sketch-text guided 3D content generation. The proposed model synergizes 3D Gaussian Splatting, ControlNet, and IP-Adapter, addressing the limitations of previous methods in terms of fidelity and consistency. Although one reviewer expressed concerns about its user study and comparison with Nerf-based methods, the authors appear to have addressed these issues in their rebuttal. Besides, the work was recognized as pioneering in incorporating 3D Gaussian Splatting into sketch-guided 3D generation, and the results are impressive. Therefore, I recommend accepting this paper.